# Functional Analysis of Viable Circulating Tumor Cells from Triple-Negative Breast Cancer Patients Using TetherChip Technology

**DOI:** 10.3390/cells12151940

**Published:** 2023-07-26

**Authors:** Vasileios Vardas, Julia A. Ju, Athina Christopoulou, Anastasia Xagara, Vassilis Georgoulias, Athanasios Kotsakis, Catherine Alix-Panabières, Stuart S. Martin, Galatea Kallergi

**Affiliations:** 1Laboratory of Biochemistry/Metastatic Signaling, Section of Genetics, Cell Biology and Development, Department of Biology, University of Patras, GR-26504 Patras, Greece; up1088956@upatras.gr; 2Department of Pharmacology, University of Maryland School of Medicine, Baltimore, MD 21201, USA; juliaju0914@gmail.com (J.A.J.); ssmartin@som.umaryland.edu (S.S.M.); 3Oncology Unit, ST Andrews General Hospital of Patras, GR-26332 Patras, Greece; athinachristo@hotmail.com; 4Laboratory of Oncology, Faculty of Medicine, School of Health Sciences, University of Thessaly, GR-41110 Larissa, Greece; xagaraa@hotmail.com (A.X.); thankotsakis@uth.gr (A.K.); 5Hellenic Oncology Research Group (HORG), GR-11526 Athens, Greece; georgulv@otenet.gr; 6Department of Medical Oncology, University General Hospital of Larissa, GR-41110 Larissa, Greece; 7Laboratory of Rare Human Circulating Cells (LCCRH), University Medical Center of Montpellier, 34295 Montpellier, France; panabieres@yahoo.fr; 8CREEC/CANECEV, MIVEGEC (CREES), Université de Montpellier, CNRS, IRD, 34090 Montpellier, France; 9European Liquid Biopsy Society (ELBS), 20246 Hamburg, Germany

**Keywords:** metastasis, immune checkpoints, EMT, circulating tumor cells, triple-negative breast cancer, vinorelbine

## Abstract

Metastasis, rather than the growth of the primary tumor, accounts for approximately 90% of breast cancer patient deaths. Microtentacles (McTNs) formation represents an important mechanism of metastasis. Triple-negative breast cancer (TNBC) is the most aggressive subtype with limited targeted therapies. The present study aimed to isolate viable circulating tumor cells (CTCs) and functionally analyze them in response to drug treatment. CTCs from 20 TNBC patients were isolated and maintained in culture for 5 days. Biomarker expression was identified by immunofluorescence staining and VyCap analysis. Vinorelbine-induced apoptosis was evaluated based on the detection of M30-positive cells. Our findings revealed that the CTC absolute number significantly increased using TetherChips analysis compared to the number of CTCs in patients’ cytospins (*p* = 0.006) providing enough tumor cells for drug evaluation. Vinorelbine treatment (1 h) on live CTCs led to a significant induction of apoptosis (*p* = 0.010). It also caused a significant reduction in Detyrosinated α-tubulin (GLU), programmed death ligand (PD-L1)-expressing CTCs (*p* < 0.001), and disruption of McTNs. In conclusion, this pilot study offers a useful protocol using TetherChip technology for functional analysis and evaluation of drug efficacy in live CTCs, providing important information for targeting metastatic dissemination at a patient-individualized level.

## 1. Introduction

The development of new treatments for breast cancer (BC) was achieved by advances in technology; however, metastasis, the primary cause of cancer-related deaths, is still not well understood and is largely incurable. Metastasis is associated with the presence of circulating tumor cells (CTCs) and disseminated tumor cells (DTCs) in peripheral blood and bone marrow, respectively [1]. The cellular processes involved in metastasis include invading the stroma, avoiding immune surveillance by inhibiting antitumor processes, altering and adapting to the tissue microenvironment, and developing resistance to treatment methods [2].

Epithelial-to-mesenchymal transition (EMT) and cytoskeletal changes are a critical component of the metastatic cascade, as it is necessary for epithelial cells to be able to enter the bloodstream, survive the shear forces of circulation, and exit into other tissues [3,4]. Microtentacles (McTNs) are tubulin-based protrusions of the plasma membrane that constitute an important mechanism for metastatic dissemination and are also associated with EMT pathways [1,5]. These structures are mechanistically distinct from traditional actin-based protrusions such as filopodia, invadopodia, and podosomes [6]. Research has shown that breast cancer cells in nonadherent environments create McTNs on their surface that help them reattach to endothelial cells and travel into the bloodstream [7]. These McTNs are supported by vimentin intermediate filaments and microtubules stabilized with post-translational modifications, such as detyrosination or acetylation of α-tubulin [8,9]. According to the results of our previous study, it was observed that CTCs could communicate with each other via thin, elongated protrusions (McTNs), which were sustained by the proteins TUB, VIM, and GLU [1]. Additionally, the fact that McTNs remain intact even after detachment indicates that VIM is co-localized with GLU within these structures, whereas cytokeratin is not [9]. Disruption of McTNs reduces metastasis of CTCs in mouse models [10,11]. A recent study demonstrated that vinorelbine, a drug that depolymerizes microtubules, showed a greater impact on reducing McTNs, reattachment, and tumor cell clustering compared to tumor cell viability [12]. These findings also suggested that vinorelbine may be a more effective first-line treatment for TNBC than previously thought, as it appears to reduce metastasis more than primary tumor growth.

The destruction of the metastatic process is important for patients’ survival [13]. Imaging studies (such as mammography, MRI, and PET-CT) are the main tools for detecting and monitoring breast cancer. They rely on the presence of tumor foci containing over 10 million tumor cells to achieve clinical detectability [12]. Due to this detection limit, patient stratification and drug development for breast cancer are based on tumor growth rather than the determinants of metastatic spread [13]. It is essential to recognize therapies that could diminish the metastatic capability of CTCs, which may be disregarded by drug development pipelines and clinical trials that are mainly concentrated on tumor growth.

BC is a complex disease including different subtypes [14]. Among these subtypes, triple-negative breast cancer (TNBC)—characterized by a lack of estrogen, progesterone, and HER2 receptors—accounts for 15% of all breast cancers and is known for its aggressive nature. However, TNBC patients currently have limited treatment options, underscoring the urgency for new biomarkers to identify potential targets for therapy. Compared to other subtypes of BC, TNBC is unique and exhibits a heterogeneous nature [15]. TNBC is also characterized by a distinct tumor microenvironment (TME) that sets it apart from other subtypes. The TME in TNBC plays a significant role in promoting cell proliferation, angiogenesis, inhibition of apoptosis, immune system suppression, and drug resistance [16]. The National Comprehensive Cancer Network (NCCN) guidelines currently recommend chemotherapy as the sole treatment option for TNBC patients. However, the guidelines do not provide specific recommendations regarding therapies that may help reduce the risk of metastasis [17]. The recent FDA approval of sacituzumab govitecan-hziy in 2020 for metastatic TNBC indicates some progress toward targeted therapies [18]. A deeper comprehension of the co-evolution between the tumor and immune system highlights the importance of adopting a holistic perspective on TNBC as an interconnected ecosystem, considering both the inherent characteristics of cancer cells and their interactions with the surrounding environment [19]. This enhanced understanding of TNBC biology has already sparked the emergence of groundbreaking targeted treatments such as PARP inhibitors, antibody–drug conjugates, and immune checkpoint inhibitors. These novel therapies are revolutionizing the field of medicine, offering promising prospects for patients at various stages of TNBC [19]. However, primary tumors cannot always provide the necessary biomarkers/companion diagnostic tests for the administration of these targeted therapies. Therefore, liquid biopsy could solve this problem by providing new insight into tumor biology. Our recent study identified PD-L1, CTLA-4, GLU, and VIM as significant biomarkers in TNBC, and their presence was linked to patients’ outcomes, offering new therapeutic opportunities for this challenging BC subtype [20]. However, there is still a requirement to improve treatment options for people with early stage TNBC in order to minimize the risk of metastasis.

The technical challenges of imaging nonadherent tumor cells are a major obstacle to comprehending how tumor cells react to the nonadherent microenvironments of metastasis, such as the bloodstream or lymphatic system. TetherChip is a microfluidic device that was engineered to prevent cell adhesion with an optically clear, thermal-crosslinked polyelectrolyte multilayer nanosurface and a terminal lipid layer that simultaneously tethers the cell membrane for improved spatial immobilization [21]. The thermal imidization process applied to the TetherChip nanosurface on commercially available microfluidic slides enables an impressive capture rate of up to 98% of tumor cells using lipid tethers [21]. Notably, the application of time-lapse microscopy reveals that the distinctive McTNs present on nonadherent tumor cells are rapidly destroyed during chemical fixation. However, when these McTNs are tethered to the TetherChip surface, their structural integrity is effectively preserved both after fixation and post-isolation from blood samples [21]. Remarkably, TetherChips exhibit exceptional stability for over 6 months, facilitating their transportation to remote locations. This microfluidic device presents a pioneering platform that goes beyond enumeration, aiming to enable functional phenotype testing in CTCs. The ultimate objective is to identify patient-specific, antimetastatic therapies [21]. In the present study, we use this novel cell tethering technique, recently developed by Professor S.S Martin’s lab at the University of Maryland [21], to isolate viable CTCs from TNBC patients and performed functional analysis for the metastatic capacity of CTCs, McTN formation, and expression of important biomarkers such as GLU, VIM, PD-L1, and CTLA-4, following (FDA-approved) vinorelbine treatment. The current study focused on TNBC not only because of the limited options for targeted therapies, but also because of the higher prevalence of McTNs in this BC subtype [7,9], which provided an opportunity to observe and investigate these structures and evaluate the efficacy of drugs targeting them. Therefore, the aim of this study in using the TetherChip technology was to evaluate the efficacy of drugs on viable CTCs within a microenvironment that simulates the nonadherent conditions of the bloodstream and to investigate the metastatic potential of these cells through the formation of McTNs. By replicating bloodstream conditions, we aimed to gain valuable insights into the behavior of CTCs. Moreover, the anchorage of free-floating CTCs in the TetherChip platform allowed the visualization of microtentacles, which cannot be observed in adherent conditions. This offers a deeper understanding of the mechanism of McTN expansion and the impact of distinct drugs on these structures. Therefore, our protocol using TetherChip technology offers valuable contributions to the understanding of CTC biology and therapeutic approaches at a personalized level.

## 2. Materials and Methods

### 2.1. Patient’s Samples and Cytospins’ Preparation

The following inclusion criteria were applied for patients’ selection: chemotherapy-naïve patients (*n* = 20) with histologically documented triple-negative breast cancer (TNBC), aged > 18 years old, with either metastatic (*n* = 7) or early stage (*n* = 13) disease. We chose patients before the initiation of any treatment cycle to avoid drug-induced alterations in the CTCs’ phenotypes. Written informed consent was obtained from all participants, and the study received approval from both the ethics and scientific committees of our institution (15/12/21-6734). To collect peripheral blood samples, 10 mL of blood was drawn from patients’ veins using EDTA as an anticoagulant. The initial 5 mL of blood was discarded to prevent contamination from skin epithelial cells during the sampling procedure. Ficoll–Hypaque density gradient was used to isolate peripheral blood mononuclear cells (PBMCs). Blood was centrifuged at 1800 rpm for 30 min at 4 °C. The blood components were separated based on their densities. Red blood cells (RBCs) and other heavy components settled at the bottom of the tube, while PBMCs remained suspended, creating a characteristic ring that was carefully isolated by pipetting under sterilized conditions. Cells were then washed twice with PBS and centrifuged at 1500 rpm for 10 min. Aliquots of 500,000 cells were centrifuged at 2000 rpm for 2 min on glass slides [22,23]. Cytospins were dried up and stored at −80 °C, as we have previously reported [23,24,25,26].

### 2.2. Cell Cultures

Breast cancer cell lines MDA-MB-231 and MDA-MB-436 were obtained from the American Type Culture Collection (ATCC; Manassas, VA, USA) and cultured in high-glucose Dulbecco’s modified eagle medium (DMEM; Thermo Fisher Scientific, Waltham, MA, USA) with 10% fetal bovine serum (FBS; PAN-Biotech, Passau, Germany) and 2 mM L-glutamine (Thermo Fisher Scientific, Waltham, MA, USA). A patient-derived colon CTC-MCC-41 cell line was obtained under a collaboration agreement between the University Hospital of Montpellier and the University of Patras and cultured in a Roswell Park Memorial Institute (RPMI; Thermo Fisher Scientific, Waltham, MA, USA) medium supplemented with 1% L-glutamine, 1% ITS (insulin, transferrin, and selenium; Thermo Fisher Scientific, Waltham, MA, USA), 10% FBS, 20 ng/mL EGF (Thermo Fisher Scientific, Waltham, MA, USA), and 10 ng/mL FGF (Thermo Fisher Scientific, Waltham, MA, USA). Finally, cells (4 million) from TNBC patients were cultured in RPMI medium supplemented with 1% L-glutamine, 1% ITS, 10% FBS, 20 ng/mL EGF, and 10 ng/mL FGF. Cells from TNBC patients were cultured for 4–5 days. This approach aimed to deplete as many PBMCs as possible to reduce blood-cell-related noise in the TetherChip. The cell culture also enriched the CTC population and provided the critical number of cancer cells to study drug efficacy in the TetherChip. The cells were cultured in a humid environment containing 5% carbon dioxide (CO_2_) and 95% air. For subculturing, a solution of 0.25% trypsin (Thermo Fisher Scientific, Waltham, MA, USA) and 5 mM EDTA (Thermo Fisher Scientific, Waltham, MA, USA) was used.

### 2.3. MTT Assay

For MTT assay, triplicate cell-culture experiments were performed using MDA-MB-231 and MDA-MB-436 (15 × 10^3^ cells/well) and CTC-MCC-41 (20 × 10^3^ cells/well) in 48-well plates. The cells were incubated overnight for attachment (MDA-MB-231 and MDA-MB-436) and then starved for 18–24 h. Vinorelbine (10 μM; Abcam, Cambridge, MA, USA (#71486)), diluted in 0.9% DMSO (AppliChem GmbH, Darmstadt, Germany), was added in triplicates and incubated for 1 h and 24 h at 5% CO_2_ at 37 °C. Thereafter, samples were incubated with 3-[4,5-dimethylthiazol-2-yl]-2,5-diphenyltratrazolium bromide (MTT; Sigma Chemical Co., St. Louis, MO, USA). After two hours, all the media including MTT solution (5 mg/mL) were removed. The residual formazan crystals were dissolved in isopropanol, and the absorbance was measured at 555 nm using a 96-well microplate reader (SynergyTM HT, Bio-Tek Instruments, Inc., Winooski, VT, USA).

For nonadherent conditions (CTC-MCC-41), 10 mg/mL poly(2-hydroxyethyl methacrylate) (Sigma Chemical Co., St. Louis, MO, USA) were used to coat the well plates.

### 2.4. Immunofluorescence Staining

Triple immunofluorescence experiments with the following combination of antibodies, CK/CD45/PD-L1, CK/CD45/CTLA-4, and CK/GLU/VIM, were performed in patients’ cytospins. Control experiments were carried out in cytospins spiked with MDA-MB-231 and MDA-MB-436 cells. To create negative controls, the corresponding primary antibody was excluded while including its secondary IgG antibody. For further characterization of isolated cells as CTCs, the cytomorphological criteria proposed by Meng et al. [27] (for example, high nuclear/cytoplasmic ratio, larger cells than white blood cells) were used. Cells were initially fixed with cold acetone, methanol 9:1, followed by 5% FBS overnight blocking. PD-L1 (1:100; Novus Biologicals, Englewood, CO, USA) was identified using anti-goat antibody labeled with Alexa Fluor 488 (Thermo Fisher Scientific, Waltham, MA, USA). For CTLA-4 detection, a CTLA-4 (F-8) Alexa Fluor 488 antibody (1:100; Santa Cruz Biotechnology, Santa Cruz, CA, USA) was employed. The anti-CD45 antibody (common leukocyte antigen), which served the purpose of excluding the potential ectopic expression of cytokeratins by hematopoietic cells, was identified using anti-CD45 conjugated with Alexa Fluor 647 (1:100; Novus Biologicals). GLU (1:400; Abcam) was detected using anti-rabbit antibody labeled with Alexa Fluor 647 (Thermo Fisher Scientific). VIM (1:100; Santa Cruz) was detected using anti-goat antibody labeled with Alexa Fluor 555 (Thermo Fisher Scientific). CK detection was achieved with the primary A45-B/B3 antibody (1:100; Amgen, Thousand Oaks, CA, USA) and its secondary anti-mouse antibody labeled with Alexa Fluor 488 (Thermo Fisher Scientific). Finally, cells were stained with 4′,6-diamidino-2-phenylindole (DAPI)-containing antifade reagent.

### 2.5. TetherChip Analysis

For immunofluorescence in TetherChips, 50,000 cells were allowed to tether for 30 min onto TetherChip, then fixed with 3.7% formaldehyde/PBS, washed, permeabilized in 0.1% Triton-X 100/PBS, blocked in 5% FBS/PBS, and incubated overnight at 4 °C. The antibody sequence was the same as in the cytospins. For McTN visualization, wheat germ agglutinin (WGA, Alexa Fluor 488 conjugate, Invitrogen, 1:100), and Hoechst 33258 (1:5000) were diluted in 1% FBS/PBS, added to each channel, and incubated for 1 h at room temperature. Based on the findings of Thompson et al. [12], vinorelbine was used in the present study to treat TNBC patients’ CTCs and their McTNs. Following 30 min for the tethering of cells, vinorelbine was added for 1 h onto TetherChip. M30 CytoDeath monoclonal antibody (Sigma Chemical Co., St. Louis, MO, USA) was used to detect apoptosis.

Cytospins and TetherChips were analyzed using the VyCAP system (VyCAP B.V., Enschede, The Netherlands) and a Leica TCS SP8 confocal microscope (Leica Microsystems, Wetzlar, Germany).

### 2.6. Statistical Analysis

Paired *t*-test analysis was used to compare the average number of CTCs detected in each patient’s cytospins, obtained after Ficoll density isolation, and the number of CTCs of the same patient located in TetherChips following 4–5 days of culture, the average number of CTCs before and after vinorelbine treatment in TetherChips, and the viability of cancer cells (MTT). Spearman analysis was used to correlate the different phenotypes in cytospins and TetherChips. A Mann–Whitney test and χ^2^ tests were used to compare the expression of PD-L1, CTLA-4, GLU, and VIM before and after 1 h vinorelbine treatment. All statistical analyses were conducted using IBM SPSS statistics version 27 software (IBM, Armonk, NY, USA). A significance level of *p* ≤ 0.05 was employed to determine statistically significant findings.

## 3. Results

### 3.1. TetherChip Analysis of Cancer Cell Lines and Live CTCs

Exploiting the fact that tethered conditions improve visualization of McTNs, we first focused on the expression of McTNs on human breast cancer cell lines (MDA-MB-231 and MDA-MB-436) and on the patient-derived colon CTC-MCC-41 cell line. The formation of McTNs was observed in all cancer cell lines (Figure 1).

### 3.2. Cytospin vs. TetherChip Analysis of TNBC Patients’ CTCs

Consequently, we compared the number of CTCs detected in cytospins after Ficoll density isolation (500,000 PBMCs per patient) with the number of CTCs in TetherChips after 4–5 days of culture (50,000 cultured PBMCs were added to TetherChips per patient). The comparison revealed a statistically significant increased number of CTCs (*p* = 0.006) located in TetherChips (Figure 2A and Table 1). Moreover, an evaluation of the CTCs’ number in TetherChips vs. cytospins among early and metastatic TNBC patients also revealed a significantly higher number of CTCs counted in TetherChips in both settings (Figure 2B). Consequently, the followed protocol with TetherChip analysis provides dramatic enrichment of CTCs per patient. The number of isolated CTCs per patient is shown in Table 1.

TNBC patients exhibited a high prevalence of McTNs in TetherChips. Interaction between two or more cancer cells through their McTNs was common, as was interaction between cancer cells and PBMCs (Figure 3).

### 3.3. Characterization of CTCs in Cytospins and in TetherChips

Focusing on the expression pattern of cytospins among TNBC patients, the expressions of PD-L1, CTLA-4, GLU, and VIM were present in 45% (9 out 20), 40% (8 out of 20), 20% (4 out of 20), and 45% (9 out of 20), respectively (Figure 4A). Regarding disease status, the expression of PD-L1 was present in 23% (3 out 13) of early TNBC vs. 87% (6 out of 7) of metastatic TNBC patients (*p* = 0.007), while the expression of CTLA-4 was present in 23% (3 out of 13) of early TNBC vs. 71% (5 out of 7) of metastatic TNBC patients (*p* = 0.035, Figure 4B). Furthermore, the expression of GLU was present in 0% (0 out 13) of early TNBC vs. 57% (4 out of 7) of metastatic TNBC patients (*p* = 0.002), while the expression of VIM was present in 46% (6 out of 13) of early TNBC vs. 43% (3 out of 7) of metastatic TNBC patients (Figure 4B). Hence, PD-L1, CTLA-4, and GLU expression was higher in metastatic compared to early TNBC patients. The frequencies of PD-L1^+^, CTLA-4^+^, GLU^+^, and VIM^+^ phenotypes among the total number of TNBC patients’ CTCs were 80%, 62%, 33%, and 88%, respectively (Figure 4C).

Considering disease status, the percentages of CTCs expressing PD-L1^+^ in early vs. metastatic disease were 50% vs. 100% (*p* = 0.018), respectively. The same percentages for the CTLA-4^+^ phenotype were 50% vs. 77%, respectively (Figure 4D). In addition, the percentages of CTCs expressing GLU^+^ in early vs. metastatic disease were 0% vs. 81% (*p* = 0.004), respectively. The same percentages for VIM^+^ phenotype were 100% vs. 69%, respectively (Figure 4D). Hence, the frequency of the phenotypes PD-L1^+^ and GLU^+^ was statistically higher in metastatic TNBC compared to early TNBC patients’ CTCs.

To investigate potential differences in the expression patterns of the examined molecules, before and after cell culture, the frequency of different phenotypes was compared in cytospins (prepared immediately after PBMCs isolation) and in TetherChips. The PD-L1^+^ phenotype was observed in 80% of CTCs in cytospins compared to 92% in TetherChips. In terms of the CTLA-4^+^ phenotype, it was found in 62% of cytospin CTCs and 41% of TetherChip CTCs. The GLU^+^ phenotype showed a frequency of 33% in cytospin CTCs and 89% in TetherChip CTCs (*p* = 0.010), whereas the GLU^−^ phenotype was present in 67% of cytospin CTCs and 11% of TetherChip CTCs (*p* = 0.010). Finally, the VIM^+^ phenotype was detected in 87% of cytospin CTCs and 98% of TetherChip CTCs (Figure 5A). Consequently, the expression of PD-L1, CTLA-4, and VIM was not statistically influenced using TetherChip analysis; however, the expression of GLU was significantly higher in TetherChips compared to cytospins. Particularly, the expression of GLU was significantly higher in TetherChips compared to cytospins among the early TNBC patients (*p* < 0.001, Figure 5B), whereas no significant differences were observed among metastatic TNBC patients’ CTCs (Figure 5C).

Representative images depicting the expression of all the different biomarkers in CTCs of TNBC patients located in TetherChips are illustrated in Figure 6.

### 3.4. Control Experiments of Vinorelbine Effects in TNBC and in CTC-41 Cell Lines

The effect of vinorelbine was first evaluated in TNBC cell lines and in the only available cell line (CTC-41) derived from CTCs isolated from a cancer patient with metastatic colon cancer. MTT analysis revealed that the viability of MDA-MB-231 (83%, *p* = 0.007) and MDA-MB-436 (87%, *p* = 0.003) was reduced after 1 h vinorelbine treatment, whereas 24 h vinorelbine treatment induced significant loss of viability of MDA-MB-231 (56%, *p* < 0.001) but less significant loss of MDA-MB-436 (70%, *p* = 0.009) (Figure 7A,B). Additionally, cell rounding and detachment of the cells was observed just after 1 h vinorelbine treatment (Figure 7D,E).

In agreement with the aforementioned results from the BC cell lines, the effect of vinorelbine in the CTC-41 cell line (cultured in nonadherent conditions) revealed that cell viability (77%) remained high but statistically different compared to control cells after 1 h vinorelbine treatment, whereas 24 h vinorelbine treatment induced significant loss of viability of CTC-41 (58%) cells (*p* < 0.001, Figure 7C).

### 3.5. Effect of Vinorelbine on PD-L1, CTLA-4, GLU, VIM Expression, and Induction of Apoptosis in MDA-MB-436 in TetherChips

To standardize the protocol for studying drug efficacy on tumor cells, using TetherChip technology, we treat the cancer cell line MDA-MB-436 with vinorelbine in a TetherChip. We used this cancer cell line for control experiments because of its superior visualization of McTNs.

Regarding the expression of the immune checkpoint molecules in MDA-MB-436, our findings revealed that the percentage of PD-L1^+^ cells before and after 1 h vinorelbine treatment was 81% and 54%, respectively (Appendix A). On the other hand, the percentage of CTLA-4^+^ cells before and after 1 h vinorelbine treatment was 95% and 77%, respectively.

Regarding the expression of EMT-related molecules (GLU and VIM), the percentage of GLU^+^ cells before and after vinorelbine treatment was 88% and 64%, respectively. Additionally, the percentage of VIM^+^ cells before and after vinorelbine treatment was 88% and 92%, respectively (Appendix A).

Finally, the study of apoptosis on these cells showed that M30^+^ cells before and after vinorelbine treatment was 5% and 37%, respectively, whereas the same percentages for M30^−^ cells were 95% and 63%, respectively.

Representative images depicting the expression of all the different biomarkers in MDA-MB-436 cells are illustrated in Appendix A.

### 3.6. Effect of Vinorelbine on PD-L1, CTLA-4, GLU, VIM Expression, and Induction of Apoptosis in TNBC Patients

Initially, to investigate the effect of vinorelbine on the total number of CTCs, we conducted a comparative analysis of the mean number of CTCs prior to and after vinorelbine administration. Our findings showed that there was a significant difference in the average number of CTCs before and after vinorelbine treatment (*p* = 0.008, Table 2).

Considering the established association between PD-L1 and CTLA-4 with disease progression, as supported by evidence from our published research [20], we performed an investigation of the expression of these immune checkpoint molecules in live CTCs isolated from the TNBC patients. Additionally, we assessed the impact of a 1 h vinorelbine treatment on the expression levels of these molecules. Our findings revealed that the percentage of PD-L1^+^ CTCs before and after vinorelbine treatment was 92% vs. 77%, respectively (*p* < 0.001), while the same percentages for PD-L1^−^ CTCs were 8% vs. 23% (*p* < 0.001), respectively, indicating a significant effect of vinorelbine treatment on PD-L1-expressing CTCs (Figure 8). Considering CTLA-4 expression, the percentage of CTLA-4^+^ CTCs before and after vinorelbine treatment was 41% and 43%, respectively. Similarly, the same percentages for CTLA-4^−^ CTCs were 59% and 57%, respectively. These findings indicated no significant effect of vinorelbine treatment on CTLA-4 expression (Figure 8).

Extending our analysis, we explored the effect of vinorelbine on EMT-related molecules (GLU and VIM). We found that the percentage of GLU^+^ CTCs before and after vinorelbine treatment was 89% and 67% (*p* < 0.001), respectively, whereas the same percentages for GLU^−^ CTCs were 11% and 33% (*p* < 0.001), respectively, demonstrating that vinorelbine treatment causes a significant reduction in GLU expression (Figure 8). Additionally, the percentage of VIM^+^ CTCs before and after vinorelbine treatment was 98% and 89%, respectively, whereas the same percentages for VIM^−^ CTCs were 2% and 11%, respectively; however, this finding did not reach a statistical significance (Figure 8).

Upon establishing the impact of vinorelbine on the expression levels of PD-L1, CTLA-4, GLU, and VIM, we proceeded to investigate whether vinorelbine could elicit apoptotic responses in the live CTCs of TNBC patients. We found that the percentage of M30^+^ CTCs before vs. after 1 h vinorelbine treatment was 16% and 30% (*p* < 0.010), respectively, whereas the same percentages for M30^−^ CTCs were 84% and 70% (*p* < 0.043), respectively (Figure 8). These findings provide compelling evidence that exposure to vinorelbine is capable of inducing apoptotic responses in the CTCs of TNBC patients.

### 3.7. Vinorelbine Effects on McTNs

Furthermore, we determined the effect of 1 h vinorelbine treatment on McTNs of TNBC patients’ CTCs. Our experiments revealed an impressive disruption of these structures in CTCs after treatment (Figure 9).

## 4. Discussion

It is widely accepted, as recently published in Nature Reviews Clinical Oncology [13], that there is a great need to find treatments that can reduce metastasis and to move away from over-reliance on response evaluation criteria for solid tumors (RECIST), which are based primarily on the imaging measurement of tumor growth rather than metastatic features. Furthermore, results from a clinical trial in 2017 indicated that CTCs can be detected in the bloodstream an average of 6 months before metastasis is observed on a PET/CT scan [28]. Metastatic breast cancer is still an incurable disease, and metastasis is the main cause of death in these patients. Recent improvements in liquid biopsy techniques have demonstrated the prognostic value of CTCs, indicating the importance of CTCs analysis for clinical practice [29]. Findings from a recent study, using TN human cancer cell lines (MDA-MB-231 and MDA-MB-436), showed that the microtubule-depolymerizing drug, vinorelbine, reduced the metastatic phenotypes of MCTNs, reattachment, and tumor cell clustering rather than tumor cell viability [12]. These findings also suggested that vinorelbine may be a more effective treatment for TNBC than previously thought, as it appears to reduce metastasis more than primary tumor growth [12].

Taking these into account in the current study, we focused on patients with TNBC because it is the most aggressive phenotype, it rapidly progresses to metastasis, and it is characterized by a limitation of targeted therapies. We evaluated the metastatic capacity of live CTCs from TNBC patients under conditions similar to the bloodstream microenvironment (nonadherent) using the TetherChip device. More specifically, we isolated live CTCs and characterized them in relation to their metastatic capacity, McTN formation, and expression of potentially important biomarkers such as GLU, VIM, PDL-1, and CTLA-4, following (FDA-approved) vinorelbine treatment.

Although the most aggressive subset of CTCs that can survive in the bloodstream and form metastasis are largely unknown, it has been shown that metastatic BC cell lines adopt a morphologic cellular reattachment phenotype, which is associated with the presence of McTNs [9,11,30]. A higher McTN number is found in more invasive BC cell lines [9]. Furthermore, the molecular mechanisms that are linked to McTNs are correlated with a greater risk of metastasis and a poorer prognosis for the patient [8]. Given the important role of McTNs in metastasis, we focused on the expression of these cytoskeletal structures in human TNBC cell lines (MDA-MB-231 and MDA-MB-436), as well as in the CTC-line CTC-MCC-41 and in live CTCs isolated from TNBC patients. Our results demonstrated the formation of McTNs across all cancer cell lines (Figure 1) and in the CTCs of TNBC patients (Figure 3). McTNs have been previously documented and analyzed in MDA-MB-231 and MDA-MB-436 cell lines [12]; however, this is the first study showing McTN formation in a CTC line (CTC-MCC-41) obtained from a colon cancer patient before the initiation of any treatment. Our analysis also revealed intracellular communication through McTNs formation both among cancer cells in CTC clusters and between CTCs and PBMCs (Figure 3). This finding agrees with our former studies showing that McTNs on CTCs isolated from BC patients appeared to play a role in facilitating communication between CTCs, as well as interactions between CTCs and surrounding blood cells [1].

To gain a better comprehension of the biology of CTCs, especially the metastasis-initiator CTCs, functional assays are urgently needed. Unfortunately, the current CTC capture methods have low yield, making it difficult to obtain enough viable CTCs for these types of functional assays. To address this issue, it is necessary to increase the number of CTCs and develop more efficient CTC capture methods [31]. Technologies like CTC-Chip, CellSearch, and other similar systems are designed to capture CTCs based on specific phenotypes or surface markers expressed on tumor cells [32,33]. These systems use antibodies or antibody-coated surfaces to target and capture CTCs with specific characteristics, such as the expression of epithelial cell adhesion molecule (EpCAM) or other epithelial markers. On the other hand, the TetherChip technology allows the capture and observation of CTCs without relying on specific markers. It aims to study the metastatic potential of cancer cells by facilitating the visualization of McTNs. Therefore, TetherChip utilizes a lipid-based tethering approach to immobilize CTCs, enabling their examination and analysis [21]. This technology has the potential to capture and visualize different phenotypic subpopulations of CTCs without being limited to specific surface markers. It also provides a more unbiased approach to study drug efficacy on CTCs’ subpopulations. In our study, we cultured PBMCs, isolated by Ficoll–Hypaque density gradient, for 4–5 days and subsequently analyzed them in TetherChips. The comparison between the number of CTCs found in each patient’s sample directly after blood collection (cytospins) and the number of CTCs of the same patients analyzed in TetherChips showed that there was a significantly increased number of CTCs in the TetherChip device, suggesting that this protocol could improve the step of CTC enrichment. Subsequently, a functional assay can be applied to get a profile of viable CTCs and their response to treatment.

Moreover, we compared the phenotypic characteristics of patients’ CTCs in cytospins and TetherChips. Our results revealed that, in cytospins, the expression levels of PD-L1, CTLA-4, and GLU were notably increased in the metastatic setting of TNBC patients (Figure 4). This finding is in line with the results of our recent research [20]. Regarding the comparison between TetherChips and cytospins, there was no statistically significant differences regarding different phenotypes, implying that our protocol does not influence the physiology of CTCs. The only phenotype that was increased in TetherChips was the GLU-positive one. Particularly, GLU expression in CTCs was significantly increased (*p* = 0.01) in TetherChips compared to in cytospin preparations in both early and total TNBC patients (Figure 5). This could be attributed to the fact that GLU participates in McTN formation and aids cancer cell migration and invasion by providing structural support [34]. Additionally, the augmented stability of microtubules supported by GLU may contribute to the resistance of cancer cells to chemotherapy [34].

In this study, we also aimed to investigate the effect of vinorelbine treatment on the viability of CTCs based on previous findings on human cancer cell lines [12]. Therefore, the TNBC cell lines (MDA-MB-231 and MDA-MB-436), as well as the patient-derived colon CTC-MCC-41 line, were treated with vinorelbine for 1 h and 24 h. Our findings revealed a substantial reduction in cell viability across all three cancer cell lines, underscoring the profound impact of vinorelbine treatment on their viability (Figure 7).

Consequently, we assessed the effect of vinorelbine on patients’ CTCs (in TetherChips) and on their distinct phenotypes that we have recently shown to be relevant to patients’ outcomes, such as PD-L1, CTLA-4, GLU, and VIM [20] (Figure 8). The presented findings demonstrate a noteworthy reduction in PD-L1 and GLU expression following vinorelbine treatment (*p* < 0.001). High levels of PD-L1 expression have been associated with worse OS of early TNBC and NSCLC patients and with shorter PFS of metastatic breast cancer (MBC) patients [35,36,37,38,39]. On the other hand, high GLU expression has been linked to significantly decreased OS and PFS of NSCLC and BC patients, respectively. Its overexpression has also been shown in McTNs, associated with EMT pathways [1,35]. Based on our data, vinorelbine treatment decreases the expression of these two biomarkers (PD-L1 and GLU) in CTCs, implying that vinorelbine could be an effective treatment by inhibiting metastasis and improve patient outcomes.

One additional question was whether vinorelbine treatment could induce apoptosis in CTCs. Our results indicated a significant increase in the percentage of M30^+^ CTCs following vinorelbine treatment, suggesting that vinorelbine induced apoptosis in CTCs of TNBC patients, reinforcing the importance of this drug for targeting metastatic spread (Figure 8). Furthermore, our results have shown that vinorelbine disturbs McTNs in patients’ CTCs (Figure 9) after one hour of treatment, in agreement with previous studies, showing that vinorelbine reduces the metastatic phenotypes of McTNs in MDA-MB-231 cells [12]. There are different strategies to disrupt McTNs such as kinesin inhibitors or curcumin treatment [3,40]. Furthermore, ionomycin and thapsigargin cause a sudden rise in cytoplasmic Ca^2+^ levels, which suppress the McTNs in the MDA-MB-231 and MDA-MB-436 metastatic breast cancer cell lines [41].

The small number of patients included in this study and the exclusive focus on TNBC patients is a limitation of this study. Expanding the sample size and including other cancer types would provide a more comprehensive understanding of microtentacles and drug response across different malignancies. Furthermore, the investigation of more cytokines and adhesion molecules in relation to McTNs and drug efficacy could unravel the complex interactions in the bloodstream. Furthermore, the in vitro experimental approach does not fully replicate the complexity of the in vivo tumor microenvironment. Future research incorporating in vivo models would help bridge this gap and validate the clinical relevance of our findings. However, besides these limitations, to the best of our knowledge, this is the first study testing drugs to inhibit metastatic potential of CTCs by reducing microtentacle formation using live CTCs from TNBC patients.

## 5. Conclusions

Our study highlights the importance of the functional analysis of CTCs using TetherChips technology as a valuable tool for real-time investigation of drug efficacy on isolated, cultured CTCs. Our results also revealed that vinorelbine can elicit rapid changes in critical biomarkers such as PD-L1 and GLU. It also induced the apoptosis of patients’ CTCs and disturbed the cytoskeletal organization of McTNs.

## Figures and Tables

**Figure 1 cells-12-01940-f001:**
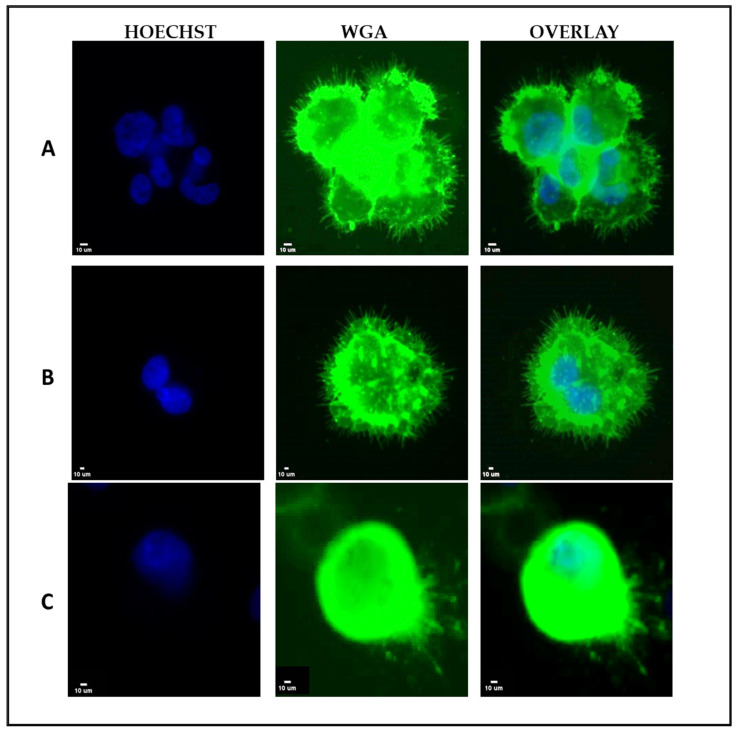
Human cancer cell lines, MDA-MB-436 and MDA-MB-231, and patient-derived colon CTC-MCC-41 cell line stained with wheat germ agglutinin (WGA) membrane dye in TetherChip. Representative VyCAP images of (**A**) tethered MDA-MB-436 cells expressing McTNs, (**B**) tethered MDA-MB-231 cells expressing McTNs, and (**C**) tethered CTC-MCC-41 cells expressing McTNs. Scale bar = 10 μm.

**Figure 2 cells-12-01940-f002:**
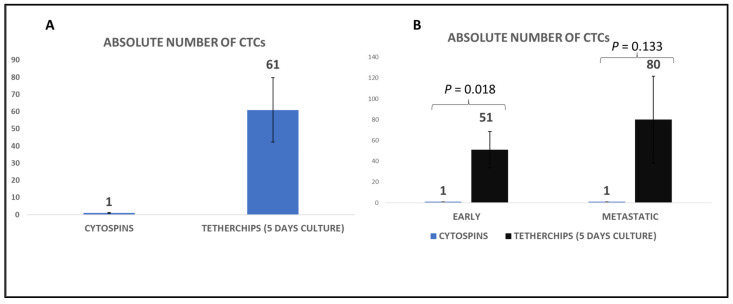
Average number of CTCs in cytospins versus TetherChips. (**A**) Comparison of the number of CTCs found in each patient’s sample isolated after the blood collection (cytospins) and the number of CTCs of the same patients located in TetherChips after 4–5 days of culture; (**B**) average number of CTCs found in early and metastatic patients in cytospins vs. TetherChips. Data are shown as mean ± SE.

**Figure 3 cells-12-01940-f003:**
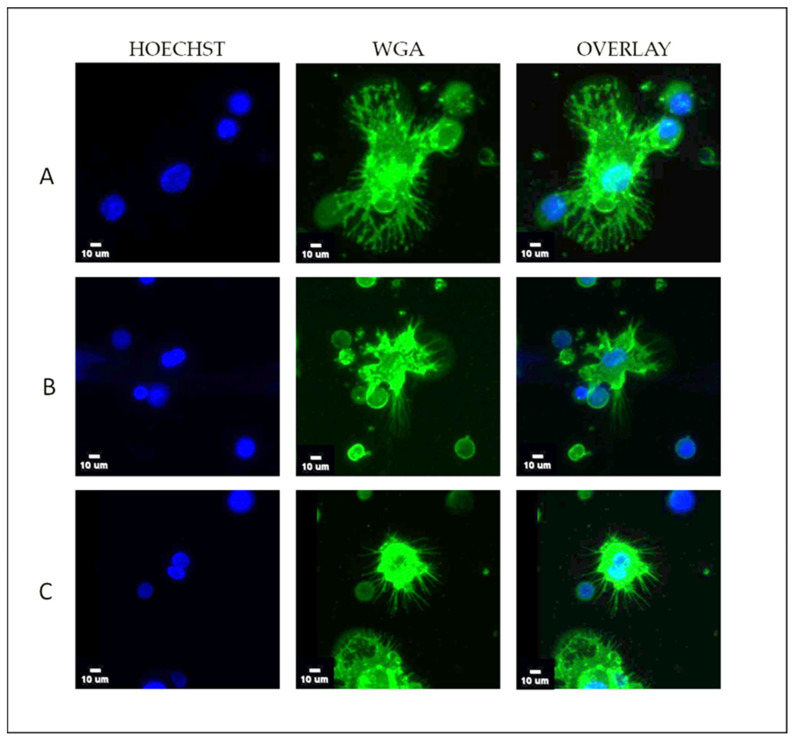
Formation of McTNs in TNBC patients’ CTCs stained with wheat germ agglutinin (WGA) membrane dye and added in TetherChips. Representative VyCAP images of (**A**) tethered CTCs of TNBC patient expressing McTNs and interacting with two PBMCs, (**B**) tethered CTCs of TNBC patient expressing McTNs and interacting with one PBMC, and (**C**) tethered CTCs of TNBC patient expressing McTNs. Scale bar = 10 μm.

**Figure 4 cells-12-01940-f004:**
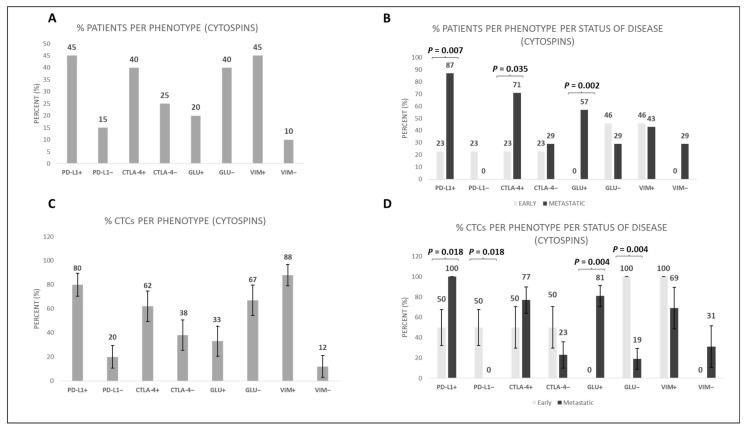
PD-L1, CTLA-4, GLU, and VIM expression in TNBC patients’ CTCs. (**A**) Percentage of TNBC patients with all the different phenotypes; (**B**) percentage of early and metastatic TNBC patients with the corresponding CTC phenotypes; (**C**) percentage of CTCs with all the corresponding phenotypes in TNBC patients; (**D**) percentage of CTCs with all the corresponding phenotypes in early and metastatic TNBC patients.

**Figure 5 cells-12-01940-f005:**
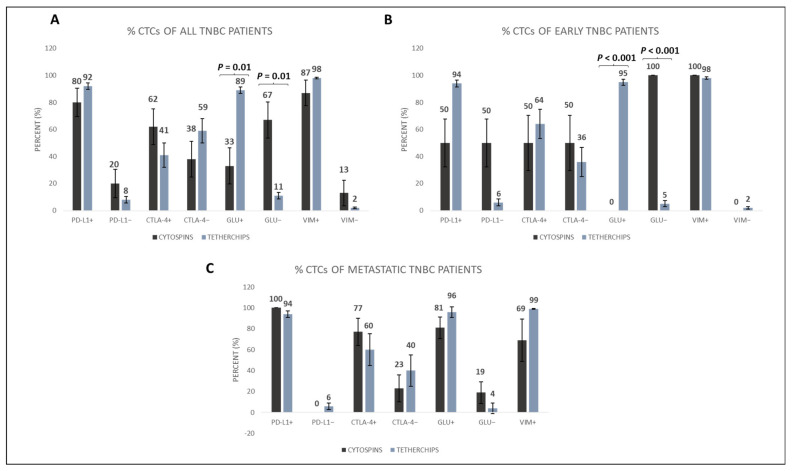
PD-L1, CTLA-4, GLU, and VIM expression in TNBC patients’ CTCs located in cytospins versus TetherChips. (**A**) Percentage of CTCs with all the corresponding phenotypes in the total TNBC patients in cytospins versus TetherChips. (**B**) Percentage of CTCs with all the corresponding phenotypes in the early TNBC patients in cytospins versus TetherChips. (**C**) Percentage of CTCs with all the corresponding phenotypes in the metastatic TNBC patients in cytospins versus TetherChips.

**Figure 6 cells-12-01940-f006:**
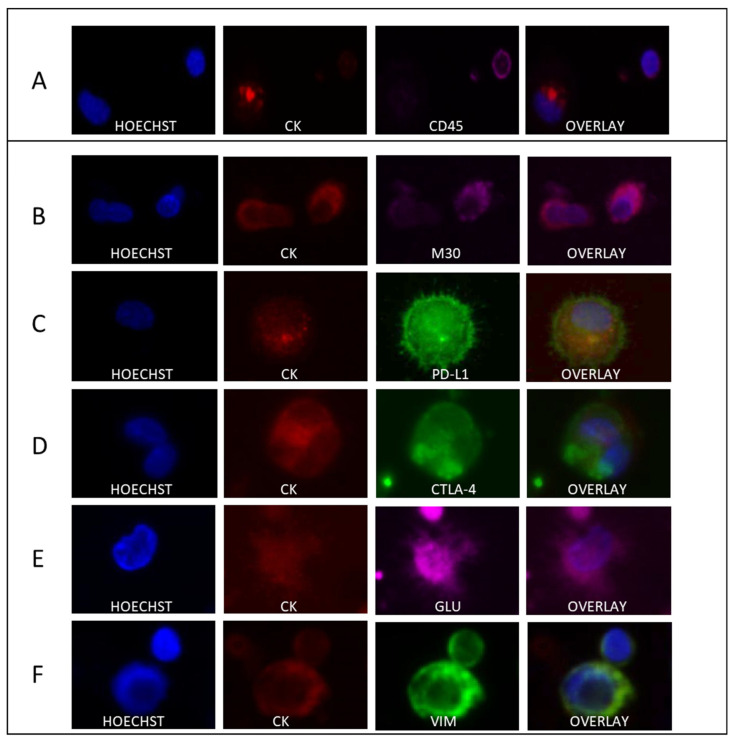
Expression of all the biomarkers in TNBC patients’ CTCs. The first column represents nuclei stained with Hoechst; the second column represents cells expressing CK, the third cells expressing CD45, M30 CytoDeath, PD-L1, CTLA-4, GLU, and VIM; the fourth represents the overlay of the three channels. Representative panels of cells with (**A**) expression of CK and CD45, (**B**) expression of M30 CytoDeath (apoptosis), (**C**) expression of PD-L1, (**D**) expression of CTLA-4, (**E**) expression of GLU, and (**F**) expression of VIM are shown.

**Figure 7 cells-12-01940-f007:**
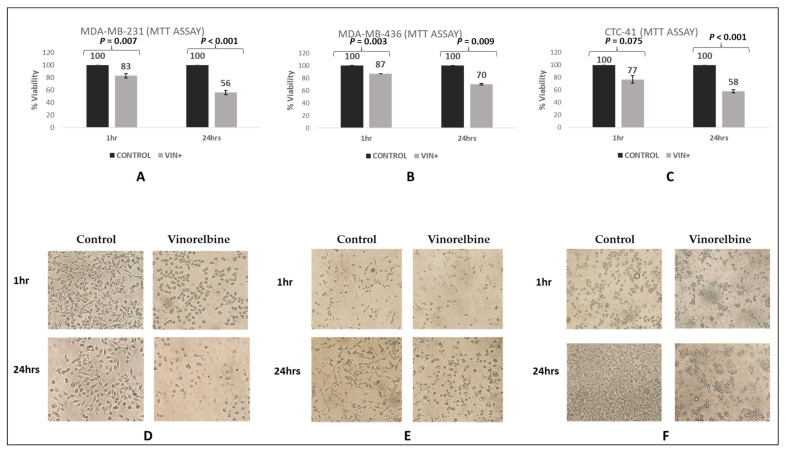
Cell viability after vinorelbine treatment. (**A**) MDA-MB-231 cells treated with vinorelbine (10 μM) for 1 h and 24 h. (**B**) MDA-MB-436 cells treated with vinorelbine (10 μM) for 1 h and 24 h. (**C**) CTC-MCC-41 cells treated with vinorelbine (10 μM) for 1 h and 24 h. Data are shown as mean ± SE, *n* = 3. Representative brightfield images were taken for each condition using a Nikon Eclipse Ti2-E inverted microscope at 10× magnification. (**D**) MDA-MB-231 treated with vinorelbine (10 μM) for 1 h and 24 h. (**E**) MDA-MB-436 treated with vinorelbine (10 μM) for 1 h and 24 h. (**F**) CTC-MCC-41 treated with vinorelbine (10 μM) for 1 h and 24 h.

**Figure 8 cells-12-01940-f008:**
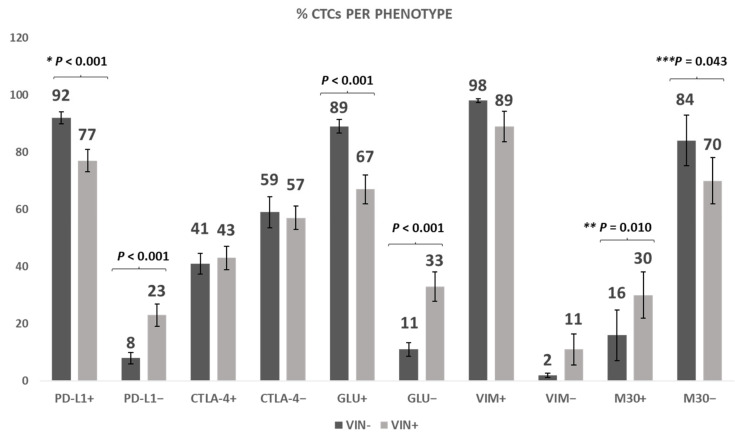
Expression of the examined biomarkers in CTCs of 20 TNBC patients following vinorelbine treatment. Percentage of CTCs with the corresponding phenotypes in early and metastatic TNBC patients before and after vinorelbine treatment (10 μΜ) for 1 h (* *p* < 0.001, ** *p* = 0.010, *** *p* = 0.043). Data are shown as mean ± SE.

**Figure 9 cells-12-01940-f009:**
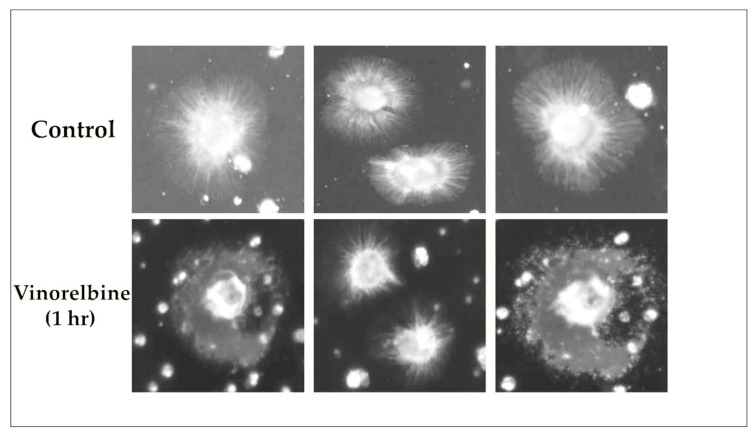
TNBC patients’ CTCs stained with wheat germ agglutinin (WGA) membrane dye, detached, and suspended in media containing vehicle (0.1% DMSO) or vinorelbine (10 μΜ) for 1 h. Representative VyCAP images of tethered CTCs of a TNBC patient with McTNs before and after the effect of vinorelbine (10 μΜ) for 1 h.

**Table 1 cells-12-01940-t001:** The average number of CTCs in cytospins vs. TetherChips. Data are shown as mean ± SE.

Patients	Status of Disease	Cytospins	TetherChip (5 Days’ Culture)
1	Early	0	111 ± 0.5
2	Early	2 ± 0.3	15 ± 1
3	Metastatic	1 ± 0.7	306 ± 25
4	Early	2 ± 1.5	31 ± 6
5	Early	1 ± 0.5	16 ± 1
6	Early	0	9 ± 2
7	Early	1 ± 0.5	231 ± 37
8	Metastatic	3 ± 0.5	32 ± 10
9	Metastatic	1 ± 0.3	17 ± 4
10	Metastatic	1 ± 0.5	186 ± 16
11	Early	1 ± 0.3	41 ± 11
12	Metastatic	1 ± 0.3	9 ± 2
13	Early	1 ± 0.3	77 ± 25
14	Early	1 ± 0.3	6 ± 1
15	Early	1 ± 0.3	3 ± 0.3
16	Early	0	13 ± 3
17	Early	1 ± 0.3	97 ± 9
18	Early	1 ± 0.5	7 ± 1
19	Metastatic	1 ± 0.3	3 ± 0.3
20	Metastatic	1 ± 0.7	5 ± 3

**Table 2 cells-12-01940-t002:** The average number of CTCs located in TetherChip prior to and after vinorelbine administration. Data are shown as mean ± SE.

TNBC Patients	Average (before Vinorelbine)	Average (after Vinorelbine)
1	111 ± 0.5	70 ± 3
2	15 ± 1	14 ± 2
3	306 ± 25	247 ± 22
4	31 ± 6	13 ± 5
5	16 ± 1	8 ± 1
6	9 ± 2	5 ± 1
7	231 ± 37	260 ± 13
8	32 ± 10	7 ± 1
9	17 ± 4	8 ± 3
10	186 ± 16	115 ± 15
11	41 ± 11	22 ± 0.3
12	9 ± 2	9 ± 4
13	77 ± 25	44 ± 0.4
14	6 ± 1	4 ± 1
15	3 ± 0.3	3 ± 1
16	13 ± 3	12 ± 2
17	97 ± 9	51 ± 18
18	7 ± 1	3 ± 0.3
19	3 ± 0.3	4 ± 1
20	5 ± 3	3 ± 1

## Data Availability

Data presented in this study are available upon request from the corresponding authors.

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
