# Peer review of "Functional Analysis of Viable Circulating Tumor Cells from Triple-Negative Breast Cancer Patients Using TetherChip Technology"

_cells, 2023, doi:10.3390/cells12151940_

Round 1
Reviewer 1 Report
"The manuscript authored by Vardas and colleagues titled 'Functional analysis of viable circulating tumor cells from triple negative breast cancer patients using TetherChip technology' is an intriguing contribution to the field of personalized medicine. The methods presented in this study hold significant potential for the development of treatments aimed at reducing cancer metastasis and improving patient survival. The manuscript introduces a novel technology called TetherChip, which enables the enrichment of circulating tumor cells (CTCs) for further analysis.
One notable characteristic of metastatic cancer cells is the presence of microtentacles, which are associated with an increased risk of metastasis and a poor prognosis. The authors demonstrate that treating enriched metastatic cancer cells with the drug Vinorelbine effectively reduces the metastatic phenotypes of tumor cells, including tumor cell reattachment and clustering. These findings suggest that Vinorelbine holds promise as an effective treatment for triple-negative breast cancer.
The manuscript is well-written and easily comprehensible, making it a valuable addition to your journal. It significantly contributes to the understanding of CTCs and their potential role in personalized medicine. However, I do have a question regarding the experimental design presented in the study.
The authors chose to culture Ficoll density-isolated peripheral blood mononuclear cells (PBMCs) from patient blood samples for 4-5 days before subjecting them to TetherChip analysis. I am curious about the rationale behind this step. Is it possible to directly use freshly isolated PBMCs for TetherChip enrichment? My concern is that the culturing process for 4-5 days might induce changes in metastatic cancer cells, potentially affecting the experimental outcomes."
Author Response
REVIEWER 1:
The manuscript authored by Vardas and colleagues titled ‘Functional analysis of viable circulating tumor cells from triple negative breast cancer patients using TetherChip technology’; is an intriguing contribution to the field of personalized medicine. The methods presented in this study hold significant potential for the development of treatments aimed at reducing cancer metastasis and improving patient survival. The manuscript introduces a novel technology called TetherChip, which enables the enrichment of circulating tumor cells (CTCs) for further analysis.
One notable characteristic of metastatic cancer cells is the presence of microtentacles, which are associated with an increased risk of metastasis and a poor prognosis. The authors demonstrate that treating enriched metastatic cancer cells with the drug Vinorelbine effectively reduces the metastatic phenotypes of tumor cells, including tumor cell reattachment and clustering. These findings suggest that Vinorelbine holds promise as an effective treatment for triple-negative breast cancer.
The manuscript is well-written and easily comprehensible, making it a valuable addition to your journal. It significantly contributes to the understanding of CTCs and their potential role in personalized medicine. However, I do have a question regarding the experimental design presented in the study.
- The authors chose to culture Ficoll density-isolated peripheral blood mononuclear cells (PBMCs) from patient blood samples for 4-5 days before subjecting them to TetherChip analysis. I am curious about the rationale behind this step. Is it possible to directly use freshly isolated PBMCs for TetherChip enrichment? My concern is that the culturing process for 4-5 days might induce changes in metastatic cancer cells, potentially affecting the experimental outcomes.
RE: We would like to thank the reviewer for the positive feedback and the kind comments on our study. His remarks are truly appreciated, and they further motivate us to expand this work to other types of cancer.
Regarding the reviewer’s comment. The rationale for culturing PBMCs for 4-5 days was to achieve the specific goal of our study, which was to evaluate drug efficacy on CTCs and on McTNs formation. To achieve this target, we needed to deplete as many PBMCs as possible, to reduce blood cell-related noise in the TetherChip. We also wanted to enrich the CTCs population to have a critical number of cancer cells for further studying drug efficacy. This rationale behind the 4-5 days of cell culture is now included in the "Materials & Methods" section, lines 173-178.
Furthermore, we understand the importance of maintaining the integrity of the cellular characteristics throughout the experimental protocol and the reviewer’s comment was also our concern. Therefore, we compared the expression profile of common biomarkers in Cytospins (immediately after PBMCs isolation) and in TetherChips. According to our findings which are referred to in the “Results” section (lines 304-315), our protocol did not influence the physiology of CTCs regarding the examined biomarkers. The only phenotype that was increased in TetherChips was the GLU-positive CTCs which was expected since GLU supports McTNs formation.
Thank you again for your valuable feedback.
Sincerely,
Galatea Kallergi

Reviewer 2 Report
This paper focuses on the functional analysis of viable circulating tumor cells (CTCs) from triple-negative breast cancer (TNBC) patients using TetherChip technology. The introduction provides background information on metastasis, the role of circulating tumor cells, and the importance of studying the metastatic process. It also discusses the significance of microtentacles in metastatic dissemination and highlights the potential of Vinorelbine, a drug that depolymerizes microtubules, in reducing McTNs and metastasis. The paper introduces TetherChip as a promising technology.
Comments:
-Clarify the objectives: The introduction would benefit from clearly stating the objectives or research questions of the study. What specific aspects of CTCs and metastasis are being investigated using TetherChip technology?
-Provide more context: While the paper mentions the significance of TNBC and the limitations of current treatment options, it would be helpful to provide more context on the challenges faced in TNBC treatment and the potential impact of identifying new biomarkers and therapeutic targets.
-Sample choice: Why TNBC was chosen to be investigated in such technic and not other type of breast cancer. Limited options of targeted therapy in TNBC as in comparison with in HER2+ cannot be act as sole reason. This may also be elaborated more. Behavior in vivo!! Is it different than other subtypes (in vitro).
Expand on TetherChip technology: The description of TetherChip is relatively brief. It would be beneficial to provide more details about how the technology works, its advantages over other methods, and its potential applications in studying CTCs and metastasis.
-How the pts were selected.
-Were these pts were selected randomly or was there any certain criteria.
-More details and clarify on how CTCs were isolated after isolated by Ficoll and centrifugation
-Language and clarity: Some sentences are long and complex, which can make the text difficult to understand. Simplifying the language and breaking down complex ideas into shorter, clearer sentences would enhance readability.
-Limitations of the study to be mentioned a.e. In vitro model is not absolutely applicable in vivo which is to be discussed. Cytokines and other adhesion molecules to be mentioned and discussed. Other comparator technologies to be discussed more in the discussion part. Challenges of TetherChip technology.
Minor:
Metastatic breast cancer is still an untreatable disease…
To be revised, I think authors mean incurable because it can be treated and pts can survive years.
Thank you. This is indeed a good project and will benefit from some modifications.
Author Response
REVIEWER 2:
This paper focuses on the functional analysis of viable circulating tumor cells (CTCs) from triple-negative breast cancer (TNBC) patients using TetherChip technology. The introduction provides background information on metastasis, the role of circulating tumor cells, and the importance of studying the metastatic process. It also discusses the significance of microtentacles in metastatic dissemination and highlights the potential of Vinorelbine, a drug that depolymerizes microtubules, in reducing McTNs and metastasis. The paper introduces TetherChip as a promising technology.
COMMENTS
- Clarify the objectives: The introduction would benefit from clearly stating the objectives or research questions of the study. What specific aspects of CTCs and metastasis are being investigated using TetherChip technology?
RE: We thank the reviewer for the valuable feedback and fruitful comments. Our aim in using the TetherChip technology was to evaluate the efficacy of drugs on viable CTCs within a simulated bloodstream microenvironment and to investigate the metastatic potential of these cells through the formation of McTNs. By replicating bloodstream conditions, we aimed to gain valuable insights into the behavior of CTCs. Moreover, the anchorage of free-floating CTCs in TetherChip platform allowed the visualization of microtentacles which cannot be observed in adherent conditions. This offers a deeper understanding of the mechanism of McTNs expansion and the impact of distinct drugs on these structures. Therefore, we believe our protocol using TetherChip technology offers valuable contributions to the understanding of CTC biology and therapeutic approaches at a personalized level.
According to the reviewer’s suggestion, we have now revised the "introduction" section (lines 131-140) to provide a more comprehensive explanation of our research objectives, using the TetherChip technology.
- Provide more context: While the paper mentions the significance of TNBC and the limitations of current treatment options, it would be helpful to provide more context on the challenges faced in TNBC treatment and the potential impact of identifying new biomarkers and therapeutic targets.
RE: We appreciate the Reviewer's valuable suggestion, regarding the need for additional context on the challenges encountered in TNBC and the potential impact of identifying new biomarkers and therapeutic targets. In response, we have made relevant changes to the revised version of our manuscript.
Specifically, we have expanded the introduction in the manuscript to provide a more comprehensive context on the role of tumor biology and microenvironment in TNBC. This can be found in lines 83-86 of the revised version. By highlighting the significance of the tumor microenvironment, we aimed to underscore the complexities and unique characteristics of TNBC, which further emphasize the importance of identifying new biomarkers and therapeutic targets in this subtype.
We have also included a section emphasizing the importance of a deeper comprehension of TNBC biology and the discovery of novel biomarkers and therapeutic targets (lines 91-102). This addition underscores the potential impact of our research and its implications for advancing the field of TNBC treatment and management.
We believe that these additions clarify the importance of liquid biopsy and the necessity of new biomarkers for TNBC patients. Therefore, we appreciate the advice of the Reviewer which we believe improved the quality and relevance of our manuscript.
- Sample choice: Why TNBC was chosen to be investigated in such technic and not other type of breast cancer. Limited options of targeted therapy in TNBC as in comparison with in HER2+ cannot be act as sole reason. This may also be elaborated more. Behavior in vivo!! Is it different than other subtypes (in vitro).
RE: Indeed, the choice to focus on TNBC was motivated, not only by the limited options for targeted therapies compared to other subtypes, but also by the higher prevalence of microtentacles (McTNs) in TNBC. Particularly, previous studies from Pr. Stuart Martin’s lab have shown that McTNs are more common in aggressive cancer cells, such as TNBC cell lines (1,2). TetherChip was specifically designed to facilitate the observation of McTNs, therefore choosing TN subtype was the best choice to test this new technology in clinical samples.
Furthermore, we took inspiration from the work of Thompson et al. 2022, (3) which demonstrated the effect of Vinorelbine on McTNs in TNBC cell lines. Building upon these findings, we aimed to extend the study to TNBC patients, thereby providing valuable insights into the response of McTNs to treatment in real clinical samples.
In conclusion, by choosing TNBC as our research focus, we aimed to address the pressing need for effective therapeutic strategies in this aggressive subtype, while also leveraging the higher occurrence of McTNs to enhance our understanding of their biology and assess the impact of targeted interventions.
We have now added an explanation of the cohort choice in the introduction section (lines 127-131).
- Expand on TetherChip technology: The description of TetherChip is relatively brief. It would be beneficial to provide more details about how the technology works, its advantages over other methods, and its potential applications in studying CTCs and metastasis.
RE: We greatly appreciate the reviewer’s valuable comment. Recognizing the importance of providing more details on the TetherChip technology, we provided more details to the revised manuscript.
Specifically, we have included a more comprehensive description of the TetherChip technology in the introduction section, spanning from lines 112 to 122. By incorporating this information, we aim to offer readers a clearer understanding of this innovative platform. This additional context will enhance the reader's comprehension of the experimental methodology and its implications for our research outcomes.
We believe that these additions strengthen the manuscript by providing the necessary background and technical insights into the TetherChip technology.
- How the pts were selected.
Were these pts were selected randomly or was there any certain criteria.
More details and clarify on how CTCs were isolated after isolated by Ficoll and centrifugation.
RE: Patients’ selection was according to specific criteria. Patients were excluded from the study if they meet any of the following criteria: a. Age < 18 years, b. Patients without histologically documented triple-negative breast cancer (TNBC), c. Patients who had received at least one cycle of therapy, and d. Patients without a signed informed consent form. We chose patients before the initiation of any treatment cycle to avoid drug-induced alterations in CTCs’ phenotypes. This information is now given in Materials & Methods section (lines 142-146).
Furthermore, all the detailed steps from the isolation by Ficoll–Hypaque density are given in lines 153-159.
- Language and clarity: Some sentences are long and complex, which can make the text difficult to understand. Simplifying the language and breaking down complex ideas into shorter, clearer sentences would enhance readability.
RE: Thank you for bringing that to our attention. We have carefully reviewed the manuscript and made the corresponding changes e.g.: in lines 304-397 to simplify the sentences and improve their clarity.
- Limitations of the study to be mentioned a.e. In vitro model is not absolutely applicable in vivo which is to be discussed. Cytokines and other adhesion molecules to be mentioned and discussed. Other comparator technologies to be discussed more in the discussion part.
Challenges of TetherChip technology.
RE: Thank you for the valuable comment. We agree that comparing the TetherChip with other relevant technologies and addressing the study limitations, would improve the manuscript. We have taken this suggestion into consideration and have made the necessary additions.
Specifically, we have included a section in the discussion (lines 472-483) that compares the TetherChip to other technologies such as CTC-chip and CellSearch® system. This comparison provides insights into each technology's advantages, limitations, and unique features, thereby highlighting the distinctive contributions of the TetherChip to the field.
Furthermore, we have incorporated a section in the discussion (lines 533-541) regarding the limitations of our study. By addressing these limitations, we provide a balanced interpretation of our findings and encourage future research to further address these aspects.
- Minor: Metastatic breast cancer is still an untreatable disease…
To be revised, I think authors mean incurable because it can be treated and pts can survive years.
RE: Thank you for pointing out the inappropriate use of the word "untreatable." We have now made the necessary correction by replacing it with the term "incurable."
I remain at your disposal for any further information.
Sincerely,
Galatea Kallergi

Round 2
Reviewer 1 Report
Revised version meet my expectations. Hence recommend for publication in the present form.
Reviewer 2 Report
Authors addressed all my comments. Thank you